# The Complete Mitogenomes of Three Grasshopper Species with Special Notes on the Phylogenetic Positions of Some Related Genera

**DOI:** 10.3390/insects14010085

**Published:** 2023-01-13

**Authors:** Chulin Zhang, Benyong Mao, Hanqiang Wang, Li Dai, Yuan Huang, Zhilin Chen, Jianhua Huang

**Affiliations:** 1Key Laboratory of Cultivation and Protection for Non-Wood Forest Trees (Central South University of Forestry and Technology), Ministry of Education, Changsha 410004, China; 2Guangxi Key Laboratory of Rare and Endangered Animal Ecology, Guangxi Normal University, Guilin 541004, China; 3Key Laboratory of Forest Bio-Resources and Integrated Pest Management for Higher Education in Hunan Province, Central South University of Forestry and Technology, Changsha 410004, China; 4College of Agriculture and Biological Science, Dali University, Dali 671003, China; 5Shanghai Entomological Museum, Chinese Academy of Sciences, Shanghai 200032, China; 6College of Life Sciences, Shaanxi Normal University, Xi’an 710119, China

**Keywords:** Acrididae, *Longzhouacris*, *Ranacris*, *Conophyma*, mitogenome, phylogenetic position

## Abstract

**Simple Summary:**

The complete mitogenomes of three grasshopper species were sequenced and annotated. The phylogenetic positions of the genera *Emeiacris* and *Choroedocus* are clarified based on both complete mitogenome and morphological evidences. The results show that *Emeiacris* consistently has the closest relationship with the genus *Paratonkinacris* of the subfamily Melanoplinae, and *Choroedocus* has the closest relationship with the genus *Shirakiacris* of the subfamily Eyprepocnemidinae, respectively. In addition, the genera *Conophymacris* and *Xiangelilacris*, as well as *Ranacris* and *Menglacris*, are two pairs of the closest relatives, but their phylogenetic positions need further study to clarify.

**Abstract:**

Clarifying phylogenetic position and reconstructing robust phylogeny of groups using various evidences are an eternal theme for taxonomy and systematics. In this study, the complete mitogenomes of *Longzhouacris mirabilis*, *Ranacris albicornis*, and *Conophyma zhaosuensis* were sequenced using next-generation sequencing (NGS), and the characteristics of the mitogenomes are presented briefly. The mitogenomes of the three species are all circular molecules with total lengths of 16,164 bp, 15,720 bp, and 16,190 bp, respectively. The gene structures and orders, as well as the characteristics of the mitogenomes, are similar to those of other published mitogenomes in Caelifera. The phylogeny of the main subfamilies of Acrididae with prosternal process was reconstructed using a selected dataset of mitogenome sequences under maximum likelihood (ML) and Bayesian inference (BI) frameworks. The results showed that the genus *Emeiacris* consistently fell into the subfamily Melanoplinae rather than Oxyinae, and the genus *Choroedocus* had the closest relationship with *Shirackiacris* of the subfamily Eyprepocnemidinae in both phylogenetic trees deduced from mitogenome protein coding genes (PCGs). This finding is entirely consistent with the morphological characters, which indicate that *Emeiacris* belongs to Melanoplinae and *Choroedocus* belongs to Eyprepocnemidinae. In addition, the genera *Conophymacris* and *Xiangelilacris*, as well as *Ranacris* and *Menglacris*, are two pairs of the closest relatives, but their phylogenetic positions need further study to clarify.

## 1. Introduction

The grasshopper family Acrididae is not only the largest group within Orthoptera, but also contains many species with tremendous economic importance [1]. Reconstructing a robust phylogeny will promote our understanding of the interesting biology and evolutionary patterns within this family, such as character and behaviour evolution [2,3,4,5]. In recent years, an increasing number of studies have been carried out based on molecular evidence to resolve phylogenetic problems at different taxonomic scales of Acrididae [6,7,8,9,10], and the complete mitogenome data has demonstrated great potential in this field [3,4,5,11,12,13,14,15,16,17,18,19,20,21,22,23,24,25,26,27,28,29,30]. Therefore, studies of the mitogenome for various purposes have always been a hot topic [4]. Since paraphyly is rampant across many subfamilies of Acrididae [17], as well as other groups of Orthoptera, such as Pyrgomorphidae, and so on [28], there is still a long way for us to go to resolve such problems. 

The genus *Longzhouacris* was first erected based on the type species *Longzhouacris rufipennis* You and Bi, 1983 [31],and currently includes 11 species distributed across tropical to subtropical areas [32]. Originally, *Longzhouacris* was placed under the family Catantopidae and not assigned to a definite subfamily. Later, it was considered to be a member of the subfamily Habrocneminae [32,33,34]. The subfamily Habrocneminae contains four genera in Li and Xia’s monograph [33]: *Habrocnemis*, *Longzhouacris*, *Menglacris*, and *Promesosternus*, but the genus *Promesosternus* is assigned now in the subfamily Oedipodinae in the Orthoptera Species File (OSF) [32]. 

The genus *Ranacris* was established with *Ranacris albicornis* You and Lin, 1983, as the type species [35] and consists of three known species so far [32,34,36]. Like the genus *Longzhouacris*, *Ranacris* was also placed originally under the family Catantopidae without a definite subfamily assignment [35]. However, later, the subfamily Ranacridinae was proposed to contain the single genus *Ranacris* [37]. This was followed by some later scholars [33,34,38]. Storozhenko [39] then synonymized Ranacridinae with Mesambriini, and the genus was transferred to the tribe Mesambriini of Catantopinae. Although the genera *Ranacris* and *Menglacris* are placed in two different subfamilies at present, they exhibit high similarity superficially except for the difference in the presence or absence of the tegmen, i.e., the genus *Ranacris* is apterous, but the genus *Menglacris* is micropterous with narrow scaly tegmina. In addition, the genus *Habrocnemis* is also very similar to *Ranacris* and *Menglacris*. 

The genus *Conophyma* is the largest one of the subfamily Conophyminae, with 102 known species distributed mainly in the mountainous and plateau areas from Central Asia to the Himalayas [32]. Most *Conophyma* species inhabit above the altitude of 2000 m. Conophyminae had been placed under the family Acrididae before Otte [40], but Eades [41] transferred it to the family Dericorythidae based on the comparison of the male genitalia structure. Dericorythidae lacks the arch sclerite characteristic of Acrididae. Instead, it has the distinctive pseudoarch that is easily mistaken for the true arch if dissection is not completed by opening up the spermatophore sac. However, Eades’ [41] study sampled only one species of the subfamily Conophyminae, *Plotnikovia lanigera*, and did not examine any materials of the genus *Conophyma*, the largest group of Conophyminae. 

Although the three genera mentioned above have a debatable phylogenetic position and a complicated relationship with other genera, there is no complete mitogenome data available for phylogeny inference. Similar problems are also explicit in the genera *Emeiacris*, *Choroedocus*, *Conophymacris,* and *Xiangelilacris*. *Emeiacris* is recognized as a member of the subfamily Oxyinae in OSF [32], but was placed in the subfamily Melanoplinae by Li and Xia [33], and Mao et al. [34], and this was always supported by molecular evidences [24,25,27]. *Choroedocus* is currently placed in the subfamily Catantopinae in OSF [32], but was obviously regarded as a member of the subfamily Eyprepocnemidinae in nearly all published literatures. *Conophymacris* is presently in the subfamily Conophyminae of the family Dericorythidae in OSF [32], but belongs to the subfamily in some monographs [33,34] and always has a closest relationship with *Xiangelilacris* in previous molecular studies [24,25,27]. In this study, the complete mitogenome of *Longzhouacris mirabilis*, *Ranacris albicornis*, and *Conophyma zhaosuensis* were sequenced and annotated. Additionally, the phylogeny of the grasshoppers in Acrididae with prosternal process was reconstructed using a selected dataset of mitogenome sequences of 90 species, including the three newly sequenced mitogenomes and the ones of 87 related species downloaded from GenBank (https://www.ncbi.nlm.nih.gov/ (accessed on 12 March 2022)). The phylogenetic position of some related genera are discussed in combination with morphological characters. 

## 2. Materials and Methods

### 2.1. Taxon Sampling

Three species, i.e., *Longzhouacris mirabilis*, *Ranacris albicornis*, and *Conophyma zhaosuensis*, were selected as representatives of the genera *Longzhouacris*, *Ranacris*, and *Conophyma*, respectively. Data of the materials for generating mitogenomes were: (1) *Longzhouacris mirabilis* (voucher number: mt1825), collected at Jiangjunzhai, Mangshan Nature Reserve, Yizhang County, Hunan Province, China; 112°55′56′′ E, 24°57′24′′ N; 22 August 2018; Jingxiao Gu leg.; (2) *Ranacris albicornis* (voucher number: mt1826), collected at Diding Nature Reserve, Jingxi County, Guangxi, China; 105°59′47′′ E, 23°5′44′′ N; 8 August 2010; Jianhua Huang leg. (3) *Conophyma zhaosuensis* (voucher number: mt1938), collected on the hill behind the stud farm, Wuzunbulake Township, Zhaosu County, Xinjiang Province, China; 81°11′56′′ E, 43°11′25′′ N, altitude 2141 m; on 15 August 2013; Jianhua Huang leg. The specimens were identified by the last author according to the keys to species in Li and Xia’s [33] and Maoet al.’s [34] monographs. They were preserved in 100% anhydrous ethanol and stored in a refrigerator(Thermo Fisher Scientific, Waltham, MA, USA) at −80 °C in the Insect Collection of the Central South University of Forestry and Technology.

All materials were collected under appropriate collection permits and approved ethics guidelines. The morphological terminology followed that of Uvarov [42] and Storozhenko et al. [43]. The terminology of male genitalia followed that of Woller and Song [44]. All photographs were taken using a Nikon D600 digital camera(Nikon Corp., Minato-ku, Tokyo, Japan) or Leica DFC 5500 system(Leica Microsystems Inc., Wetzlar, Germany), the stacking images were combined using Helicon Focus ver. 6.0, (Helicon Soft, Kharkiv, Ukraine) and the plates were edited in Photoshop CS (Adobe Inc., San Jose, CA, USA).

To clarify the phylogenetic positions of as many taxa as possible, we included in this analysis nearly all presently available mitogenome data of species with a prosternal process in Acrididae and Dericorythidae (Table 1), representing 2 families, 12 subfamilies, 50 genera and 88 species/subspecies in total. *Coryphistes ruricola* (MG993389, MG993390, MG993403, MG993406) in Catantopinae and *Kosciuscola tristis* (MG993402, MG993408, MG993414) in Oxyinae were not included in this analysis because they have only partial mitogenome sequences available. *Gesonula punctifrons* (MN046214) with complete mitogenome was also excluded from this analysis due to the inaccuracy of the sequence, possibly derived from the inadequate sequencing data (only 2 Gb data was generated through next-generation sequencing (NGS)) [24]. 

### 2.2. Sequencing, Assembly and Annotation

A hind femur for each sample was sent to Berry Genomics (Beijing, China) for genomic sequencing using NGS, and the remainder of the specimen was deposited as a voucher specimen at the Central South University of Forestry and Technology. Whole genomic DNA was extracted from muscle tissue of the hind femur using a modified routine phenol and chloroform method. Separate 400 bp insert libraries were created from the whole genome DNA and sequenced using the Illumina HiSeq X Ten sequencing platform. A total of 20 Gb of 150 bp paired-end (PE) reads were generated in total for each sample. Raw reads were filtered to remove reads containing adaptor contamination (>15 bp matching to the adaptor sequence), poly-Ns (>5 bp Ns), or >1% error rate (>10 bp bases with quality score < 20). The mitogenome sequence was assembled from clean reads in Mitobim (ver. 1.9.1, see https://github.com/chrishah/MITObim (accessed on 8 May 2022)) [67]. Two runs were implemented independently using the same reference with different starting points (one point is *trnI* and another is *COXI*) to improve the sequence quality of the control region. The assembled raw mitogenome sequences were primarily annotated online using the MITOS Web Server (http://mitos.bioinf.uni-leipzig.de/index.py; accessed on 20 May 2022) [68] and then checked and corrected in Geneious (ver. 8.04, see https://www.geneious.com(accessed on 5 June 2022)) [69]. The secondary structure of the RNA encoding genes predicted in MITOS were visualised and checked manually using VARNA (ver.3.93, see http://varna.lri.fr(accessed on 13 June 2022)) [70]. The three newly sequenced mitogenomes have been deposited in GenBank under accession number ON943039 for *Ranacris albicornis*, ON931612 for *Longzhouacris mirabilis*, and ON943040 for *Conophyma zhaosuensis*, respectively (Table 1).

Base composition, A−T- and G − C-skews, and codon usage were calculated in MEGA X [71]. The formulas used to calculate the skews of the composition were (A − T)/(A + T) for the A−T-skew and (G − C)/(G + C) for the G−C-skew.

### 2.3. Phylogenetic Analyses

To explore the phylogenetic position of the genera *Longzhouacris*, *Ranacris*, *Conophyma* and some related taxa, 96 complete mitogenome sequences in total, representing 85 species in Acrididae and 3 species in Dericorythidae, were selected as ingroups, and 2 species in Pamphagidae served as outgroups. The complete mitogenome dataset consists of the 13 protein coding genes (PCGs). The two rRNA genes were not used for the phylogeny inference due to their limited resolution above the genus level [27,29]. In order to involve more species of the genus *Conophyma* in our analysis, partial mitochondrial *COX1* sequences of 6 *Conophyma* species were downloaded from NCBI (Appendix A Appendix A) and the corresponding fragment was extracted from the complete mitogenome to generate a new dataset of partial *COX1* fragment.

The PCGs were aligned using macse_v2.03 (see https://bioweb.supagro.inra.fr/macse/index.php?menu=releases (accessed on 8 July 2022)) [72]. The alignments were manually optimized and concatenated into a single dataset in Phylosuite (see http://phylosuite.jushengwu.com/ (accessed on 9 July 2022)) [73]. 

The PCGs dataset was divided into 39 data blocks (13 PCGs divided into individual codon positions). Best-fit models of nucleotide evolution and best-fit partitioning schemes were selected using ModelFinder (see http://www.iqtree.org/ModelFinder/ (accessed on 9 July 2022)) [74]. The best-fitting models used for the phylogenetic analyses of the mitochondrial PCGs and partial *COX1* datasets are shown in Appendix A Appendix A, respectively.

The phylogenies were reconstructed in maximum likelihood (ML) and Bayesian inference (BI) frameworks. The ML phylogenies were reconstructed using IQ-TREE (ver. 1.6.12, see http://www.iqtree.org (accessed on 13 July 2022)) [75]. The approximately unbiased branch supportvalues were calculated using UFBoot2 [76]. The analysis was performed in W-IQ-TREE (see http://iqtree.cibiv.univie.ac.at (accessed on 13 July 2022)) [77] using the default settings. Nodes with a bootstrap percentage of at least 70% were considered well supported in the ML analyses [78]. BI analyses were accomplished in MrBayes (ver. 3.2.1, see http://morphbank.Ebc.uu.SE/mrbayes/ (accessed on 15 July 2022)) [79], with two independent runs, each with four Markov Chain Monte Carlo (MCMC) chains. The analysis was run for 1 × 10^7^ generations, sampling every 100 generations, and the first 25% of generations were discarded as burn-in, whereas the remaining samples were used to summarize the Bayesian posterior probabilities. All of the above analyses were implemented in Phylosuite (see http://phylosuite.jushengwu.com/ (accessed on 15 July 2022)) [73]. For the phylogenetic trees reconstructed from partial *COXI* dataset, the cutoffs of 0.95 posterior and 90 bootstrap were used to collapse the nodes below these cutoffs to a polytomy.

To overcome, at least partially, some of the issues in mtDNA, such as the generally high saturation and the among-lineages and/or among-sites compositional bias, the mitochondrial PCGs were translated into amino acids, and then the amino acids were used to run an ML and a Bayesian analysis via MtOrt [24], the taxa-specific amino acid substitution model for Orthoptera, and MtRev [74], the best-fit model chosen according to Bayesian Information Criterion (BIC), respectively.

## 3. Results

### 3.1. Characteristics of the Newly Sequenced Mitogenomes

The mitogenomes of *Longzhouacris mirabilis*, *Ranacris albicornis*, and *Conophyma zhaosuensis* are all circular molecules with total lengths of 16,164 bp, 15,720 bp, and 16,190 bp, respectively (Figure 1). They have the typical metazoan mitochondrial gene set consisting of 13 PCGs, 22 tRNAs, 2 rRNAs (the large and small ribosomal subunits), and a putative A + T-rich region (control region, CR). Among the 13 PCGs, 9 (*ATP6*, *ATP8*, *COX1*, *COX II*, *CYTB*, *ND2*, *ND3*, and *ND6*) are located in the J strand, and the remaining 4 (*ND1*, *ND4*, *ND4L*, and *ND5*) are located in the N strand. Among the 37 genes coded by the mitogenome, 23 genes are coded at the J strand and 14 at the N strand. The gene order of the newly sequenced mitogenomes is the same as that of other published mitogenomes in Caelifera (Figure 1). The base composition is obviously A–T-biased, with the total A + T contents ranging from 74.6% (*Conophyma zhaosuensis*), to 75.1% (*Ranacris albicornis*), to 75.2% (*Longzhouacris mirabilis*) (Appendix A Appendix A). The A–T-skews are 0.1185 (*Ranacris albicornis*), 0.125 (*Longzhouacris mirabilis*), and 0.1394 (*Conophyma zhaosuensis*), and the G–C-skews are −0.1498 (*Ranacris albicornis*), −0.1406 (*Longzhouacris mirabilis*), and −0.1216 (*Conophyma zhaosuensis*).

Most PCGs have a typical initiation codon of ATN (Table 2). However, *COX1* in *Longzhouacris mirabilis* and *Ranacris albicornis* initiates from a non-standard initiation codon of ACC, *COX1* in *Conophyma zhaosuensis* initiates from CAA, and *ATP6* in *Longzhouacris mirabilis* initiates from GTG. Seven PCGs (*ND2, COX2, COX3, ND4, ND4L, ND6*, and *CYTB*) initiated from ATG. The initiation codon ATT has the second highest frequency of usage, followed by ACC and ATA. With respect to termination codons, the majority of PCGs have a typical termination codon of TAA in most species (Table 2). The complete termination codon TAG occurs in *ND1* in all of the three species. The incomplete termination codon TA occurs only in *CYTB* of *Longzhouacris mirabilis* and *ND6* of *Conophyma zhaosuensis*. *COX1* in all of the three species, *ND4* in *Longzhouacris mirabilis*, and *ND4* and *ND5* in *Conophyma zhaosuensis* are terminated by T. 

The PCGs of the mitogenome have extremely similar codon usage pattern to other grasshoppers (Appendix A Appendix A). Among all codons of the PCGs, the most preferred codon with the highest average relative synonymous codon usage (RSCU) is UUA, which codes for Leucine and has an RSCU value of 3.98%. The next common codons are UCA (Serine) and CGA (Arginine), followed by UCU (Serine) and ACA (Threonine), with average RSCU values of 2.463%, 2.467%, 2.09%, and 1.993%, respectively, indicating a distinct codon usage bias in grasshoppers [29].

The sizes of the 22 tRNAs varies over a very small range in all the three newly sequenced mitogenomes (Appendix A Appendix A). Except for *tRNASer*-AGN lacking the DHU arm, all of the other 21 tRNAs can be folded into a typical clover structure (Appendix A Appendix A). The numbers of base mismatches in the tRNAs varies drastically among the different mismatch types in all species, but all species have similar distribution patterns of base mismatches (Table 3). The mismatch of G–U represents the majority of the total mismatches. A–A occurs only once in *trnD*. U–U occurs in *trnQ* and some other tRNAs. A–G occurs only in *trnW*. For the most frequent mismatch of G–U, it does not occur in *trnM*, *trnD*, *trnN*,and *trnW* for all three species, andthe maximum mismatch number in one tRNA is five (Table 4).

The *lrRNA* and *srRNA* are located between the *trnL1* and *trnV,* and *trnV* and A + T-rich regions, respectively. Their lengths vary between 1365–1389 bp (*lrRNA*) and 792–806 bp (*srRNA*). The control region is located between *rrnS* and *trnI*, and contains the highest proportion of A + T content ranging from 78.6 to 80.9%. The lengths of the control region vary between 860 and 1437 bp (Appendix A Appendix A).

In addition to the control region, there are also some gene intervals or base overlaps between some genes, and the maximum overlap area is between *trnL1* and *rrnL* (Appendix A Appendix A). There are nine, seven, and seven tightly aligned gene pairs without overlap or interval in the mitogenome of the three species, respectively.

### 3.2. Phylogeny

The phylogenetic trees inferred from the dataset of the 13 mitochondrial PCGs using maximum likelihood and Bayesian inference methods have an extremely consistent topology above the genus level (Figure 2, Appendix A Appendix A). At the family level, the monophylies of both Acrididae and Dericorythidae are not supported. The three species of Dericorythidae form three individual clades. *Conophymacris viridis* completely falls into Acrididae, having the closest relationship with *Xiangelilacris zhongdianensis*. *Conophyma zhaosuensis* and *Dericorys annulata* are located near the base of the trees, but do not form a single clade. *Conophyma zhaosuensis* forms a small clade with *Leptacris* sp. and *Dericorys annulata* forms an individual clade itself, located at the most outside of the ingroup. At the subfamily level, the monophylies of six subfamilies (Spathosterninae, Oxyinae, Cyrtacanthacridinae, Eyprepocnemidinae, Calliptaminae, and Coptacrinae) are retrieved usually with strong nodal support. However, the remaining subfamilies are not recovered as monophyletic. For the subfamily Melanoplinae, nearly all species cluster into an independent clade except for *Xiangelilacris zhongdianensis*, which forms a small clade with *Conophymacris viridis* of Dericorythidae, and has close relationships with Coptacrinae and *Longzhouacris mirabilis* of Habrocneminae. The two species of Habrocneminae sampled in this study, *Longzhouacris mirabilis* and *Menglacris maculata*, do not form a single clade, but fall into two distantly separated clades, one of which is *Menglacris maculata* + *Ranacris albicornis*, and the other is *Longzhouacris mirabilis* + *Conophymacris viridis* + *Xiangelilacris zhongdianensis* + Coptacrinae, with a bootstrap value of 100% or posterior probability of 0.96. The members of Hemiacridinae are divided into two distantly separated clades, with *Hieroglyphus* species having a close relationship with Spathosterninae, but *Leptacris* sp. forming a small clade with *Conophyma zhaosuensis*. The members of Catantopinae are divided into three clades: *Ranacris albicornis*, the genus *Traulia* and the typical Catantopini species. *Ranacris albicornis* is consistently most related to *Menglacris maculata*, with a bootstrap value of 100% or a posterior probability of 1. The genus *Traulia* has the closest relationship with the clade including Coptacrinae species. The clade of the typical Catantopini species forms the sister group of Cyrtacanthacridinae. 

Although the phylogeny deduced from the mitochondrial PCGs is robust, the phylogenetic trees reconstructed from the dataset of partial *COX1* fragment sequences exhibit great difference from the former (Appendix A Appendix A). The monophylies of the subfamilies Calliptaminae, Coptacrinae, Eyprepocnemidinae, and Oxyinae are no longer supported in both or at least one tree from the *COX1* dataset. The relationships among the key groups are also very different between the ML and BI trees, and the relationships among most clades are unsolved, forming a large polytomy at the base of the trees. Even the two *Hieroglyphus* species are also split into two distantly separated clades. This result indicates the extreme instability of the phylogeny reconstructed using COX1 sequence.

Despite the great difference in the topology between the trees from mitochondrial PCGs and *COX1* datasets, and the instability of the *COX1* trees, the small clades of *Ranacris albicornis* + *Menglacris maculata* and *Conophymacris viridis* + *Xiangelilacris zhongdianensis* are always robust in all trees (Figure 2, Appendix A Appendix A). The clade of the genus *Conophyma* is also robust in the *COX1* trees, but the relationship of this clade with other groups varies (Appendix A Appendix A). The position of *Longzhouacris mirabilis* also varies in the *COX1* trees (Appendix A Appendix A). It falls into the clade of the subfamily Eyprepocnemidinae in the ML tree of the *COX1* dataset (Appendix A), but forms a polytomy clade with some species of the subfamilies Hemiacridinae and Oxyinae, as well as other clades, in the BI tree of the *COX1* dataset (Appendix A Appendix A). In addition, it is noticeable that the genus *Emeiacris* always falls into the clade of the subfamily Melanoplinae and has an extremely stable close relationship with *Paratonkinacris* in all trees (Figure 2, Appendix A Appendix A). The similar case also occurs in the genus *Choroedocus*, which always forms a stable clade with *Shirakiacris* species of the subfamily Eyprepocnemidinae and has a closer relationship with the subfamily Calliptaminae than Catantopinae in the trees from the dataset of mitochondrial PCGs (Figure 2, Appendix A Appendix A). 

With a further look at the trees reconstructed from the amino acid dataset, we find that the ML tree reconstructed using the MtOrt model (Figure 3) has an extremely high similarity in the main topology with the trees deduced from the mitochondrial PCGs (Figure 2, Appendix A Appendix A), including the non-monophyly of Dericorythidae, the monophylies of the six subfamilies (Spathosterninae, Oxyinae, Cyrtacanthacridinae, Eyprepocnemidinae, Calliptaminae, and Coptacrinae), the positons of the genera *Emeiacris* and *Choroedocus*, the relationship of *Menglacris* with *Ranacris*, and that of *Conophymacris* with *Xiangelilacris*, and so on. The most important difference is that *Dericorys annulata* and *Conophyma zhaosuensis* forms an independent clade only in this tree (Figure 3). The BI tree deduced from the amino acid dataset using the MtRev model (Appendix A Appendix A) is similar to the ML tree, but the monophyly of the subfamily Calliptaminae is no longer supported, with *Peripolus nepalensis* escaping from the clade of the genus *Calliptamus*.

## 4. Discussion

### 4.1. Phylogenetic Position of the Genus Emeiacris

The genus *Emeiacris* was established with *Emeiacris maculata* Zheng, 1981 as the type species [80] and three known species so far [32]. According to the original description, *Emeiacris* is most similar to the genus *Oxyacris* of the subfamily Oxyinae, and is mainly characterized by the rounded apex of the lower knee-lobes of the hind femora (Figure 4e), the widely separated metasternal lobes (Figure 4f), and the distinct process at the lateral margin of the supra-anal plate. When *Emeiacris* was erected, it was not definitely assigned to any subfamily. Therefore, it is possible that the authors of OSF placed *Emeiacris* in the subfamily Oxyinae according to the original reference, where the closest relative of *Emeiacris* was *Oxyacris* of the subfamily Oxyinae [32]. Subsequently, *Emeiacris* was definitely placed in the subfamily Podisminae [81], and then in Melanoplinae [33]. Morphologically, the species of *Emeiacris* are extremely similar to the species of the genera *Ognevia* and *Fruhstorferiola* (Figure 4a–n). The rounded apex of the lower knee-lobes of the hind femora and the widely separated metasternal lobes are typical distinguishing characters of Melanoplinae (Figure 4e,f), but not those of Oxyinae (Figure 4o–q). In addition, the absence of the ectoapical spine in the hind tibia (Figure 4g,h), and the epiphallus not divided into two separated symmetric parts (Figure 4i–k), also disagree with the diagnostic characters of Oxyinae, but match those of Melanoplinae. In the molecular study, *Emeiacris* consistently falls into the clade of Melanoplinae and has the robust closest relationship with *Paratonkinacris* in all trees (Figure 2 and Figure 3, Appendix A Appendix A). Therefore, the genus *Emeiacris* should be considered as a member of the subfamily Melanoplinae rather than Oxyinae. 

### 4.2. Phylogenetic Position of the Genus Choroedocus

The genus *Choroedocus* was proposed by Bolívar [82] to replace the preoccupied genus name “*Demodocus* Stål, 1878” (nec *Demodocus* Guérin, 1843 in Coleoptera). *Demodocus* was proposed first as a subgenus of the genus *Calliptenus*, which was considered being most similar to the genus *Eyprepocnemis* [83], and then raised to the generic level by Brunner von Wattenwyl [84]. Kirby [85] proposed to restrict Walker’s name *Heteracris* to the genus because it was preoccupied in Coleoptera. Bolívar [82] proposed a new name, *Choroedocus*, for *Demodocus*. No matter *Demodocus*, *Heteracris*, or *Choroedocus*, they were always definitely assigned to the group Euprepocnemes [82,86], or the tribe Eyprepocnemini [87,88], or the subfamily Eyprepocnemidinae [33,34,38,85]. There are indeed a few works where *Choroedocus* is placed in the subfamily Catantopinae [81,89,90], but Catantopinae in this sense contains actually all Eyprepocnemidinae taxa. In other words, all Eyprepocnemidinae taxa are members of the subfamily Catantopinae, and there is no category of the subfamily Eyprepocnemidinae in that classification scheme. 

We do not know why the authors of OSF finally placed the genus *Choroedocus* in the subfamily Catantopinae. The most probable reason may be that the genus *Choroedocus* was once placed by Liu [91] in the family Catantopidae without assignment of the subfamily position. However, this is not strong evidence for the decision because the truth is that Liu [91] merely listed no subfamily category in his work. After examining materials of the genus *Choroedocus*, we found they highly agree with the distinguishing characters of the subfamily Eyprepocnemidinae: the pronotum with distinct lateral carina and a large, black, velvety maculation, the hind tibiae with many more spines on the external margins (Figure 5a–d), and the male genitalia structure, especially the epiphallus (Figure 5e–j), which is very similar to that of *Shirakiacris shirakii* (Figure 5k–t). Based on molecular analysis, *Choroedocus* has a robust relationship with *Shirakiacris* of the subfamily Eyprepocnemidinae (Figure 2 and Figure 3, Appendix A Appendix A). Therefore, it is more reasonable to consider *Choroedocus* as a member of the subfamily Eyprepocnemidinae.

### 4.3. Phylogenetic Relationships among Some Related Genera 

#### 4.3.1. Non-Monophyly of the Family Dericorythidae

The family name Dericorythidae was first proposed by Eades [41] according to the distinctive pseudoarch in the phallic complex. The pseudoarch found in Dericorythidae is a paired structure not connected across the midline (Figure 6j,s–t). In contrast, the arch of aedeagus rises from the median, dorsobasal region of the dorsal valves of aedeagus [44] (Figure 4m). Eades [41] thought that the presence of a well-developed arch sclerite should be treated as a crucial character in defining the family Acrididae. However, the representatives of Dericorythidae examined by Eades [41] were extremely limited, with only two species in the subfamily Dericorythinae, and one species in the subfamilies Conophyminae and Iranellinae, each, leading to a possibility that the morphological diversity was not fully represented by the limited taxon sampling. 

In this study, the 3 sampled species of Dericorythidae did not cluster into a single clade in all phylogenetic trees (Figure 2 and Figure 3, Appendix A Appendix A). *Conophymacris szechwanensis* first clusters into a clade with *Xiangelilacris zhongdianensis*, and then with *Longzhouacris mirabilis* and two species of Coptacrinae in the trees from the mitogenome dataset, showing an extremely distant relationship with the two other species of Dericorythidae (Figure 2 and Figure 3, Appendix A Appendix A). Furthermore, *Conophymacris szechwanensis* has not only a true arch rather than a pseudoarch (Figure 7j), but also an extremely different external morphology (Figure 7a–d) and geographical distribution. *Dericorys annulata* and *Conophyma zhaosuensis* are both located at the base of the trees, but do not form a single clade in most phylogenetic trees (Figure 2, Appendix A Appendix A), except in the ML tree deduced from amino acid dataset with MtOrt model, where *Dericorys annulata* and *Conophyma zhaosuensis* forms an independent clade (Figure 3). Although both *Dericorys annulata* and *Conophyma zhaosuensis* have pseudoarches in the phallic complex (Figure 6j,s,u) and similar geographical distribution region, they exhibit distinct differences in external morphology, including the general appearance and the male genitalia structure, especially the epiphallus (Figure 6e–i,p–r). 

Therefore, the family Dericorythidae is certainly not a monophyletic group, and the relationship among the family needs to be clarified by denser taxon and character sampling and more nuclear molecular markers, including both genome and transcriptome data [5]. After all, morphology may be heavily influenced by many abiotic and biotic factors [92]. Some morphological characters may have evolved independently multiple times [2,28] and may not be reliable for recognizing monophyletic groups within some higher categories [28,93]. Phylogenetic relationships between or within some groups may be clouded by many factors, such as gene tree discordance, introgression, and the gene tree anomaly zone [92]. Denser taxon sampling will reveal a wide range of variation and more efficiently improve an artificial classification [94,95,96]. 

#### 4.3.2. Phylogenetic Relationship between the Genera *Conophymacris* and *Xiangelilacris*

The genus *Conophymacris* was erected by Willemse [97] with *Conophymacris chinensis* Willemse, 1933 as the type species, and placed originally in the subfamily Catantopinae. Later, it was respectively placed in the tribe Conophymatini of Catantopinae [43,98], the subfamily Conophyminae [40], and Podisminae [33,34,81], but not assigned to a definite tribal position within the subfamily Conophyminae in OSF [32]. The genus *Xiangelilacris* was established by Zheng et al. [99] with the type species, *Xiangelilacris zhongdianensis*, as the only known species so far. This genus is most similar to the genera *Indopodisma* and *Pedopodisma*, according to the original description. Therefore, it was undoubtedly recognized as a member of the tribe Podismini of the subfamily Melanoplinae by later acridologists [34]. However, this opinion has not been supported by mitogenome evidence, and it always has a very robust close relationship with *Conophymacris* in all phylogenetic trees [27] (Figure 2 and Figure 3, Appendix A Appendix A). The clade of *Conophymacris* + *Xiangelilacris* fell into neither the subfamily Melanoplinae nor the clades of other species of the family Dericorythidae. We examined some materials of *Conophymacris szechwanensis* (Figure 7a–k) as well as the types of *Xiangelilacris zhongdianensis* (Figure 7l–s), and found that the types of *Xiangelilacris zhongdianensis* are nymphs rather than adults, with the bud of hind wings distinctly covering on that of tegmina (Figure 7r,s), and most similar to *Conophymacris* species. In addition, *Conophymacris szechwanensis* has a true arch sclerite in the phallic complex (Figure 7j), indicating its possible membership in the family Acrididae. Both *Conophymacris szechwanensis* and *Xiangelilacris zhongdianensis* have distinct lateral carinae on the pronotum and ectoapical spines in the hind tibiae, which are absent in Melanoplinae. Therefore, it is undoubted that the genera *Conophymacris* and *Xiangelilacris* have a most robust close relationship with each other, and neither of them belongs to either the subfamily Melanoplinae or the family Dericorythidae, or even the Conophyminae. Their exact position needs further comprehensive research. 

#### 4.3.3. Phylogenetic Relationship between the Genera *Menglacris*, *Ranacris* and *Longzhouacris*

The genus *Menglacris* was established by Jiang and Zheng [100], with *Menglacris maculata* Jiang and Zheng, 1994 as the type species. Although it was not assigned to a definite subfamily position originally, its membership in Habrocneminae was recognized by Li et al. [33], and followed by Mao et al. [34]. The taxonomic history and subfamily position of the genus *Ranacris* have been mentioned in the introduction section. Although these two genera are placed in different subfamilies in OSF, they display a very robust close relationship with each other in all phylogenetic trees (Figure 2 and Figure 3, Appendix A Appendix A) and an extremely high similarity in external morphology and male genital structure (Figure 8). However, they are always far away from the genus *Longzhouacris* of the subfamily Habrocneminae in all phylogenetic trees (Figure 2 and Figure 3, Appendix A Appendix A), and show distinct differences in morphology and male genital structure (Figure 8 and Figure 9). Therefore, we have decided that the genera *Menglacris* and *Ranacris* are close relatives, but their relationship with *Longzhouacris* and other related groups may be resolved only in a broader context where all members of the subfamily Habrocneminae and the tribe Mesambriini, or even more related groups, could be included in the analysis. 

### 4.4. Performance of the Mitochondrial COX1 Gene in Reconstructing Phylogeny 

A previous study has suggested that *COX1* barcode region may perform much better in phylogenetic reconstruction at genus and species ranks than at higher ranks [101]. In this study, the *COX1* barcode region extracted from the mitogenome plus that of six additional *Conophyma* species [102] was used again to test: (1) the accuracy of mitogenome sequence of *Conophyma zhaosuensis*; (2) the phylogenetic position of the genus *Conophyma* using more sampled species; and (3) its performance in reconstructing phylogeny at higher categories under the phylogenetic framework derived from the complete mitogenome dataset. The result showed that *Conophyma zhaosuensis* always fell within the same clade together with other *Conophyma* species (Appendix A Appendix A) [102], indicating the reliability or accuracy of the newly sequenced mitogenome of *Conophyma zhaosuensis*. Although the *Conophyma* species always formed a single clade in both ML and BI trees (Appendix A Appendix A), they did not cluster into a single clade with *Dericorys annulata* and *Conophymacris szechwanensis*, just as in the trees deduced from the complete mitogenome dataset (Figure 2, Appendix A Appendix A), indicating the remote relationship among these three genera. As for the performance of *COX1* gene in resolving the phylogeny at higher categories, the monophylies of most well-charactered subfamilies were not supported except for Spathosterninae and Cyrtacanthacridinae. Sometimes, even the congeneric species, such as the *Hieroglyphus* species, were split into different clades. Therefore, the mitochondrial *COX1* gene alone is not suitable for resolving phylogeny of higher categories, at least in Acridoidea, but is indeed powerful in reconstructing phylogenetic relationships among closely related species [103] despite the high error rates sometimes in individual lineages [104].

## Figures and Tables

**Figure 1 insects-14-00085-f001:**
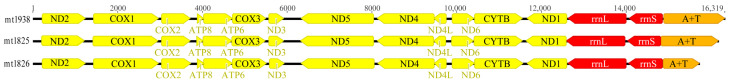
Linearised representation of gene arrangements in the three newly sequenced mitogenomes. mt1825: *Longzhouacris mirabilis*; mt1826: *Ranacris albicornis*; mt1938: *Conophyma zhaosuensis*.

**Figure 2 insects-14-00085-f002:**
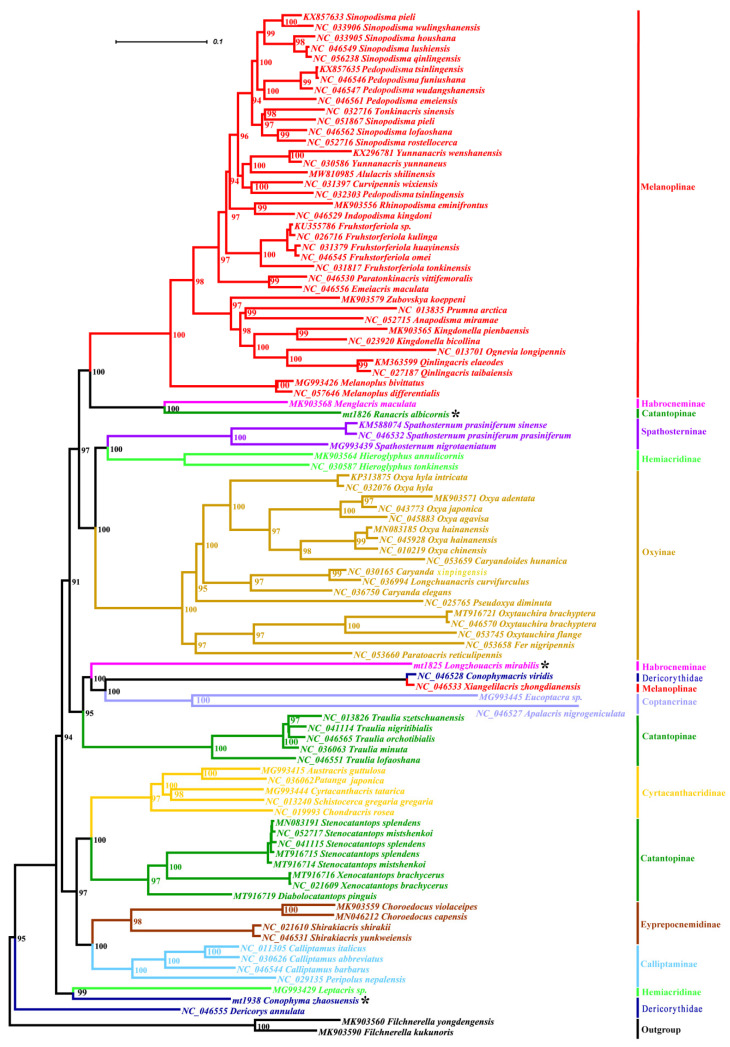
Phylogenetic tree reconstructed from the sequences of the 13 mitochondrial PCGs using maximum likelihood. The asterisk indicates the three newly sequenced species.

**Figure 3 insects-14-00085-f003:**
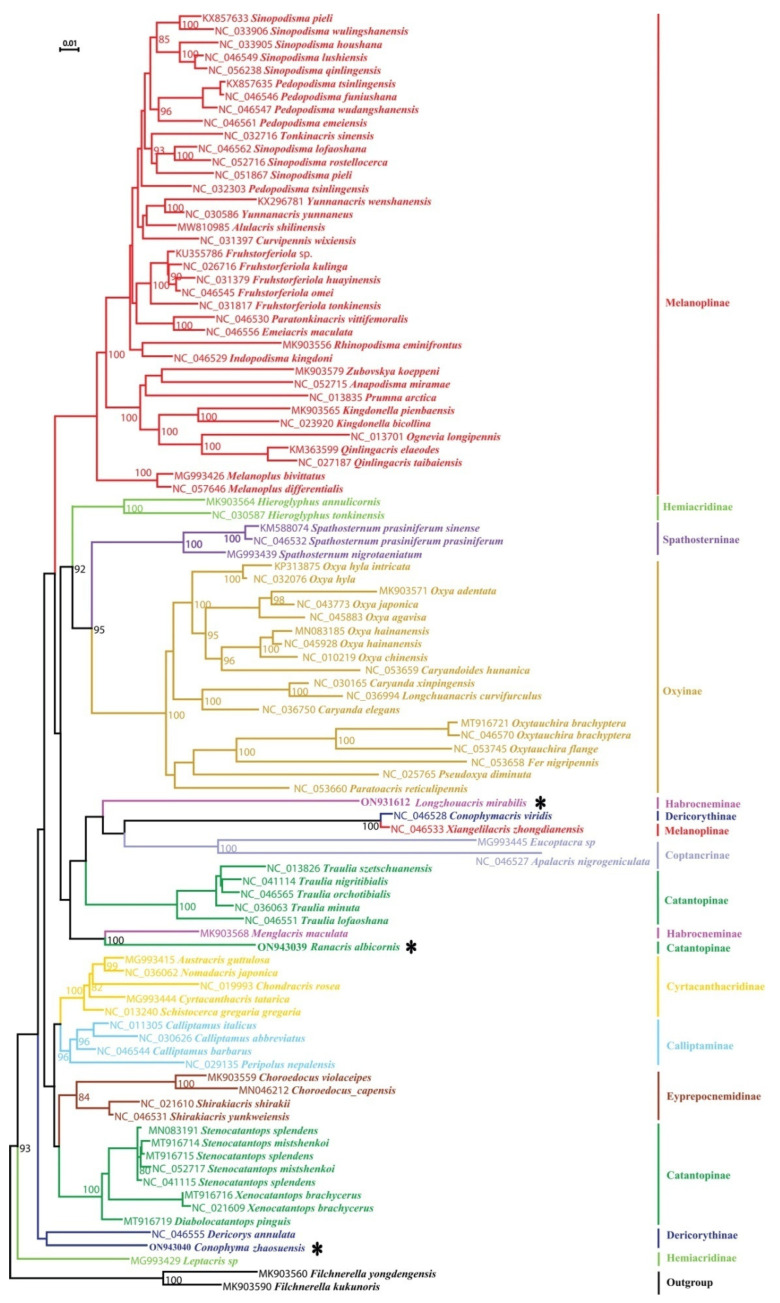
Phylogenetic tree reconstructed from amino acid sequences of the 13 mitochondrial PCGs using maximum likelihood with MtOrt model. The asterisk indicates the three newly sequenced species.

**Figure 4 insects-14-00085-f004:**
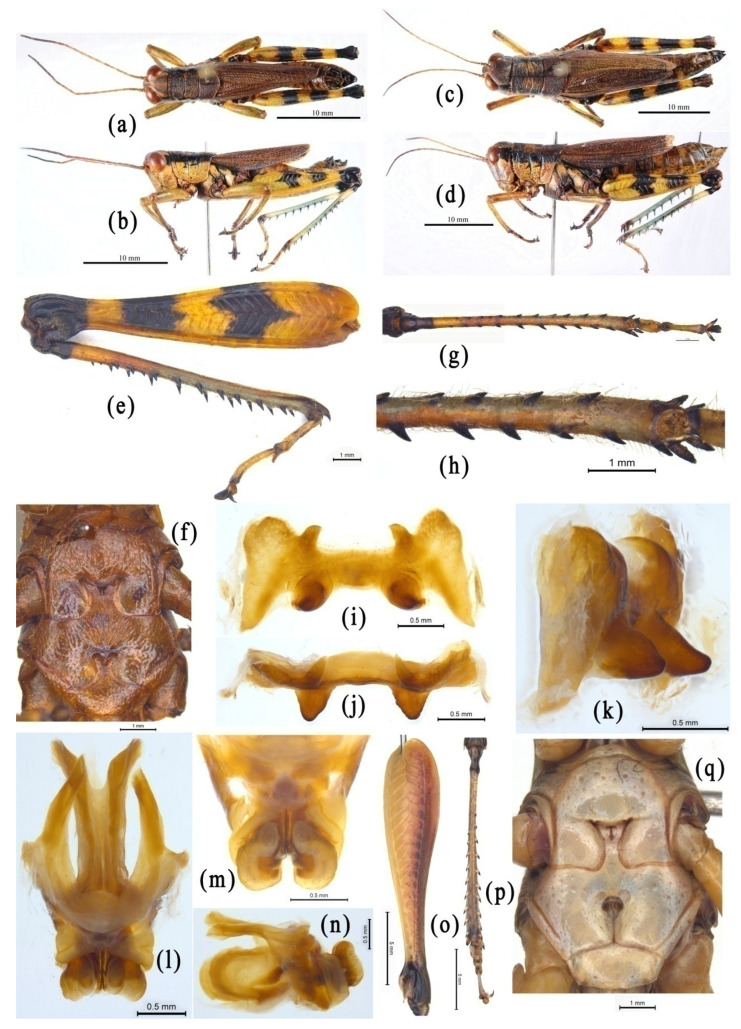
Morphological characters of *Emeiacris maculata* and *Oxya agavisa*. (**a**–**n**) *Emeiacris maculata*. (**a**,**b**) Male habitus. (**c**,**d**) Female habitus. (**e**) Right hind leg of male. (**f**) Meso- and metasterna of male. (**g**,**h**) Right hind tabia of male. (**i**–**k**) Epiphallus in dorsal, frontal and dorsolateral views. (**l**–**n**) Phallic complex in dorsal and lateral views. (**o**–**q**) *Oxya agavisa*. (**o**) left hind femur of male. (**p**) left hind tibia of male. (**q**) Meso- and metasterna of male.

**Figure 5 insects-14-00085-f005:**
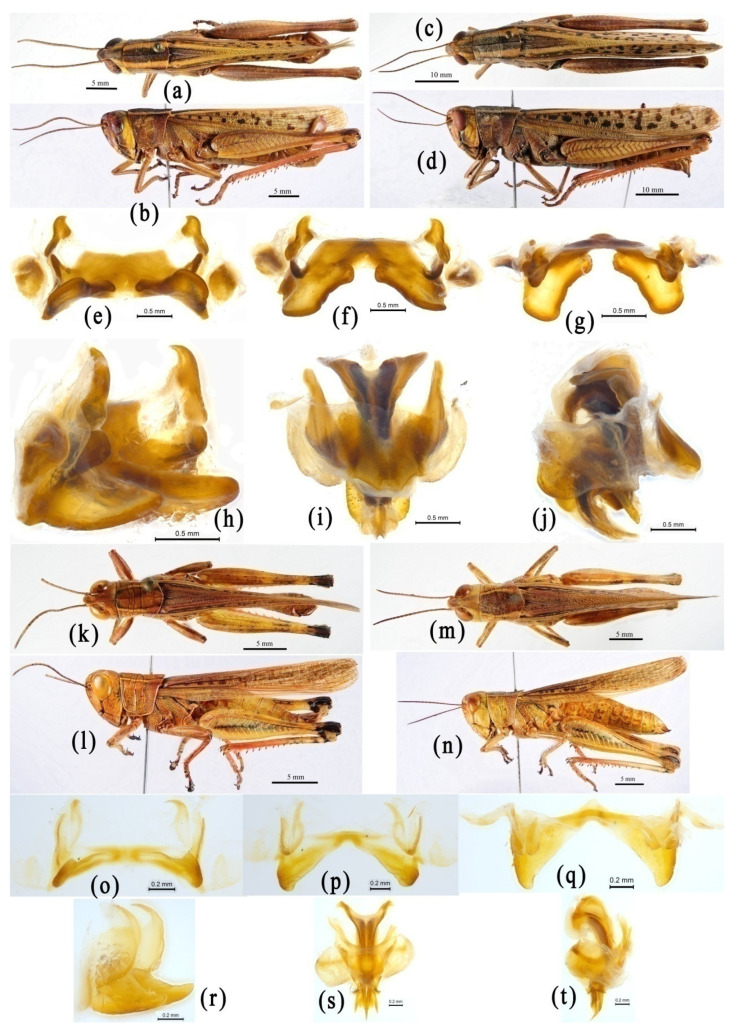
Morphological characters of *Choroedocus capensis* and *Shirakiacris shirakii*. (**a**–**j**) *Choroedocus capensis*. (**a**,**b**) Male habitus. (**c**,**d**) Female habitus. (**e**–**h**) Epiphallus in dorsal, dorsofrontal, frontal and dorsolateral views. (**i**,**j**) Phallic complex in dorsal and lateral views. (**k**–**t**) *Shirakiacris shirakii*. (**k**,**l**) Male habitus. (**m**,**n**) Female habitus. (**o**–**r**) Epiphallus in dorsal, dorsofrontal, frontal and dorsolateral views. (**s**,**t**) Phallic complex in dorsal and lateral views.

**Figure 6 insects-14-00085-f006:**
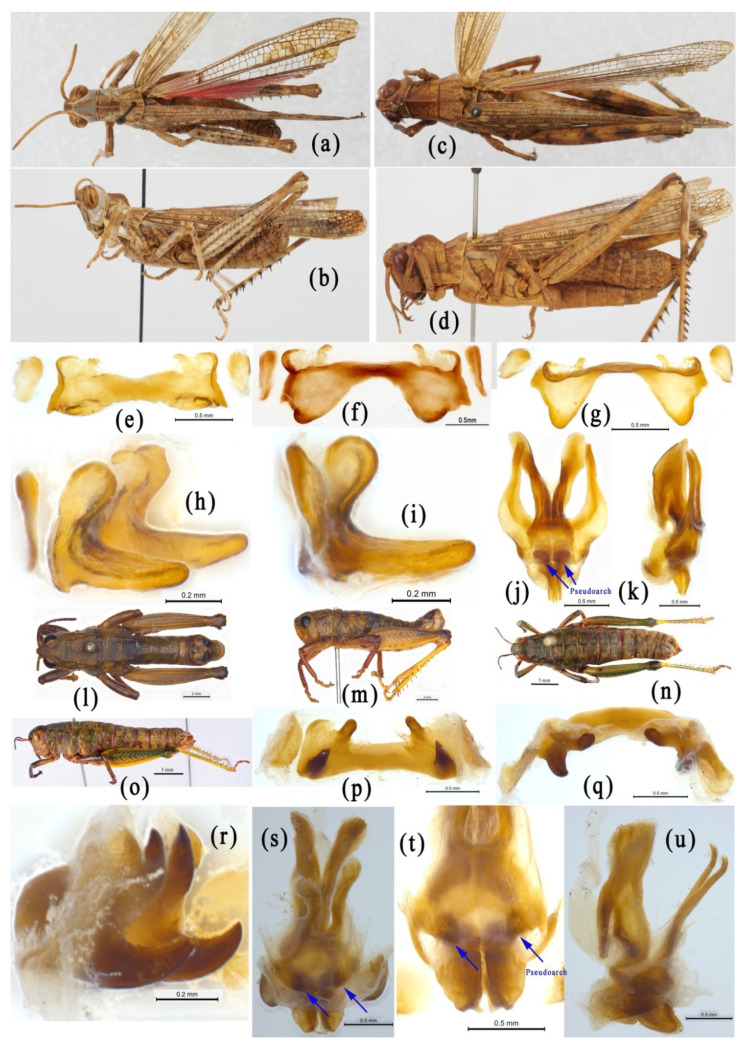
Morphological characters of *Dericorys annulata* and *Conophyma zhaosuensis*. (**a**–**k**) *Dericorys annulata*. (**a**,**b**) Male habitus. (**c**,**d**) Female habitus. (**e**–**i**) Epiphallus in dorsal, dorsofrontal, frontal, dorsolateral and lateral views. (**j**,**k**) Phallic complex in dorsal and lateral views. (**l**–**u**) *Conophyma zhaosuensis*. (**l**,**m**) Male habitus. (**n**, **o**) Female habitus. (**p**–**r**) Epiphallus in dorsal, frontal and dorsolateral views. (**s**–**u**) Phallic complex in dorsal and lateral views. The blue arrows indicate the pseudoarch.

**Figure 7 insects-14-00085-f007:**
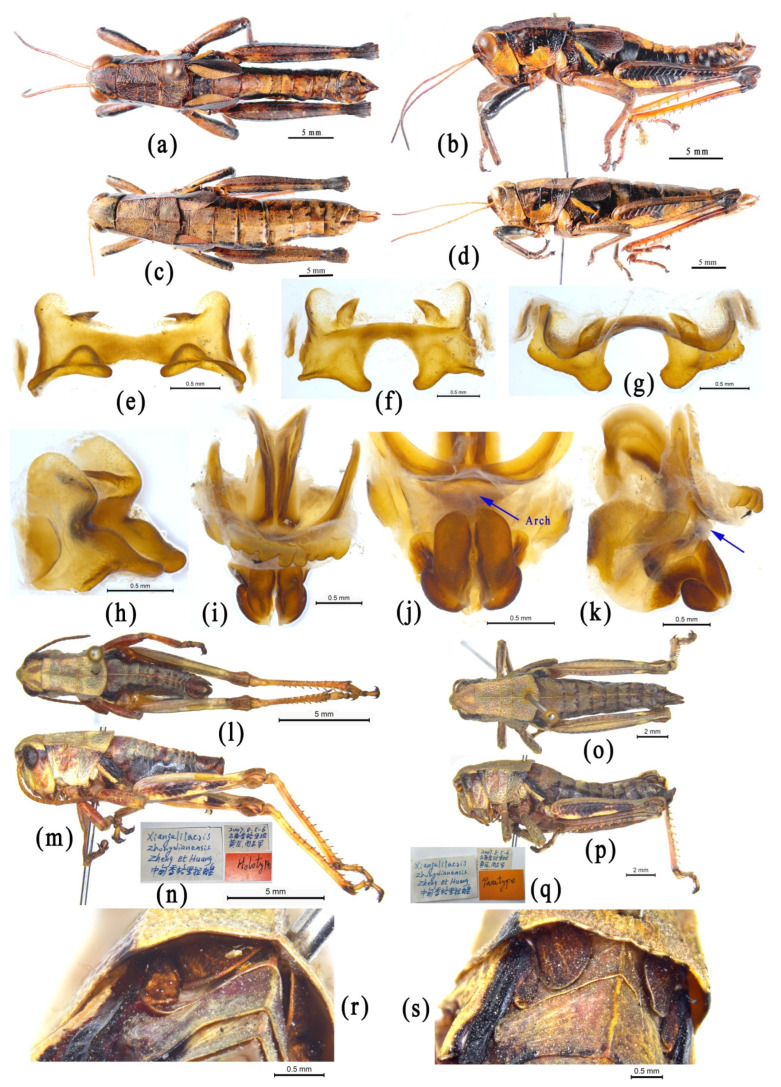
Morphological characters of *Conophymacris szechwanensis* and *Xiangelilacris zhongdianensis*. (**a**–**k**) *Conophymacris szechwanensis*. (**a**,**b**) Male habitus. (**c**,**d**) Female habitus. (**e**–**h**) Epiphallus in dorsal, dorsofrontal, frontal and dorsolateral lateral views. (**i**–**k**) Phallic complex in dorsal and lateral views. (**l**–**s**) Types of *Xiangelilacris zhongdianensis*. (**l**,**m**) Holotype male habitus. (**n**) Labels of the holotype male (the chinese character sting below the scientific name *Xiangelilacris zhongdianensis* in the left label is the chinese name for this species, and that in the upper right label is the collecting data of the holotype male: 5–6 August 2007; Xiangelila, Yunnan; Yuan Huang, Zhijun Zhou leg.). (**o**,**p**) Paratype female habitus. (**q**) Labels of the paratype female (the chinese character sting below the scientific name *Xiangelilacris zhongdianensis* in the left label is the chinese name for this species, and that in the upper right label is the collecting data of the paratype female, same as that of the holotype male). (**r**) Wing bud of the holotype male. (**s**) Wing bud of the paratype female. The blue arrows indicate the arch.

**Figure 8 insects-14-00085-f008:**
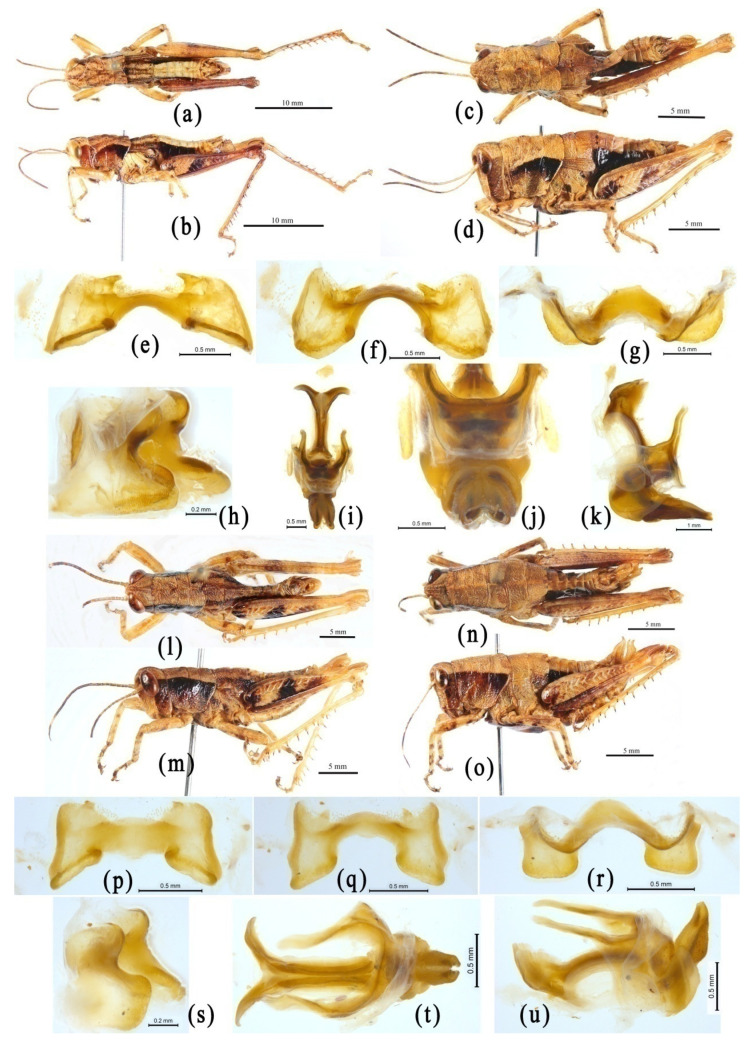
Morphological characters of *Menglacris maculata* and *Ranacris albicornis*. (**a**–**k**) *Menglacris maculata*. (**a**,**b**) Male habitus. (**c**,**d**) Female habitus. (**e**–**h**) Epiphallus in dorsal, dorsofrontal, frontal and dorsolateral views. (**i**–**k**) Phallic complex in dorsal and lateral views. (**l**–**u**) *Ranacris albicornis*. (**l**,**m**) Male habitus. (**n**,**o**) Female habitus. (**p**–**s**) Epiphallus in dorsal, dorsofrontal, frontal and dorsolateral views. (**t**–**u**) Phallic complex in dorsal and lateral views.

**Figure 9 insects-14-00085-f009:**
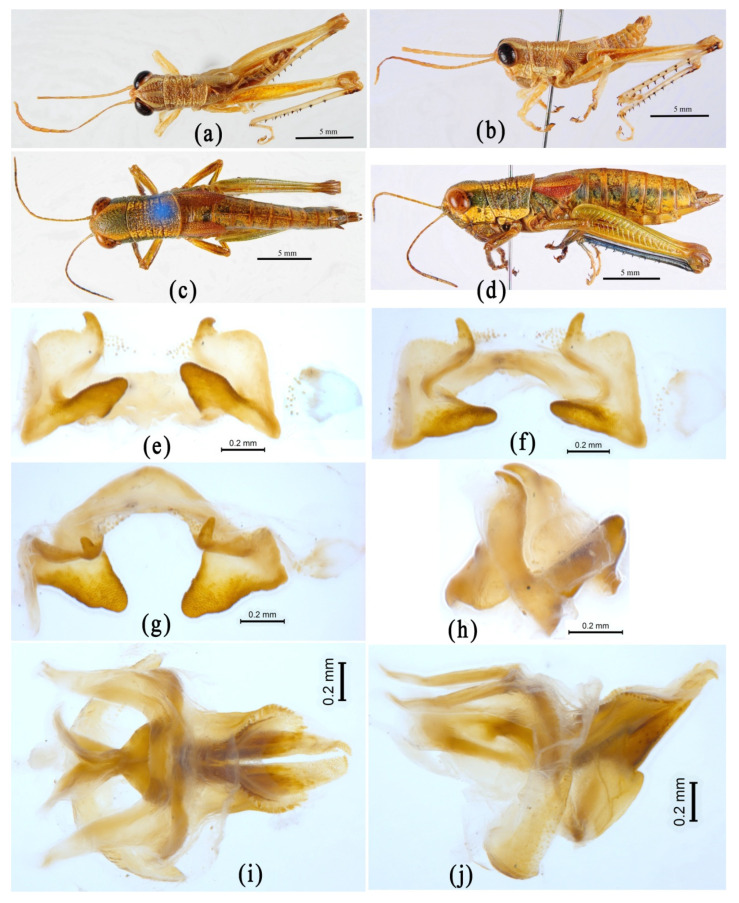
Morphological characters of *Longzhouacris mirabilis*. (**a**,**b**) Male habitus. (**c**,**d**) Female habitus. (**e**–**h**) Epiphallus in dorsal, dorsofrontal, frontal and dorsolateral views. (**i**,**j**) Phallic complex in dorsal and lateral views.

**Table 1 insects-14-00085-t001:** Accession numbers and references of the mitogenomes of the species sampled in this study.

Species	Accession Number	Reference
**Caelifera, Acridoidea, Acrididae**		
**Calliptaminae**		
*Calliptamus abbreviates*	NC_030626	Han et al. (2016) [45]
*Calliptamus barbarous*	NC_046544	Chang et al. (2020) [24]
*Calliptamus italicus*	NC_011305	Fenn et al. (2008) [11]
*Peripolus nepalensis*	NC_029135	Zhi et al. (2016) [46]
**Catantopinae**		
*Diabolocatantops pinguis*	MT916719	Zeng et al. (2021) [29]
*Ranacris albicornis*	ON943039	This study (mt1826)
*Stenocatantops mistshenkoi*	NC_052717	Chen et al. (2020) [4]
	MT916714	Zeng et al. (2021) [29]
*Stenocatantops splendens*	MN083191	Chang et al. (2020) [24]
	MT916715	Zeng et al. (2021) [29]
	NC_041115	Li et al. (2019) [20]
*Traulia lofaoshana*	NC_046551	Chang et al. (2020) [24]
*Traulia minuta*	NC_036063	Qiu et al. (2021) [47]
*Traulia nigritibialis*	NC_041114	Li et al. (2019) [20]
*Traulia orchotibialis*	NC_046565	Chang et al. (2020) [24]
*Traulia szetschuanensis*	NC_013826	Direct Submission
*Xenocatantops brachycerus*	MT916716	Zeng et al. (2021) [29]
	NC_021609	Yang J. et al. (2016) [48]
**Coptancrinae**		
*Apalacris nigrogeniculata*	NC_046527	Chang et al. (2020) [24]
*Eucoptacra* sp.	MG993445	Song et al. (2018) [17]
**Cyrtacanthacridinae**		
*Austracris guttulosa*	MG993415	Song et al. (2018) [17]
*Chondracris rosea*	NC_019993	Direct Submission
*Cyrtacanthacris tatarica*	MG993444	Song et al. (2018) [17]
*Patanga japonica*	NC_036062	Direct Submission
*Schistocerca gregaria gregaria*	NC_013240	Erler et al. (2010) [12]
**Eyprepocnemidinae**		
*Choroedocus capensis*	MN046212	Chang et al. (2020) [24]
*Choroedocus violaceipes*	MK903559	Chang et al. (2020) [24]
*Shirakiacris shirakii*	NC_021610	Direct Submission
*Shirakiacris yunkweiensis*	NC_046531	Chang et al. (2020) [24]
**Habrocneminae**		
*Longzhouacris mirabilis*	ON931612	This study (mt1825)
*Menglacris maculata*	MK903568	Chang et al. (2020) [25]
**Hemiacridinae**		
*Hieroglyphus annulicornis*	MK903564	Chang et al. (2020) [25]
*Hieroglyphus tonkinensis*	NC_030587	Chang and Huang (2016) [49]
*Leptacris* sp.	MG993429	Song et al. (2018) [17]
**Melanoplinae**		
*Alulacris shilinensis*	MW810985	Xu et al. (2021) [27]
*Anapodisma miramae*	NC_052715	Chen et al. (2020) [4]
*Curvipennis wixiensis*	NC_031397	Chen and Xu (2017) [50]
*Emeiacris maculate*	NC_046556	Chang et al. (2020) [24]
*Fruhstorferiola huayinensis*	NC_031379	Liu and Qiu (2016) [51]
*Fruhstorferiola kulinga*	NC_026716	Yang R. et al. (2016) [52]
*Fruhstorferiola omei*	NC_046545	Chang et al. (2020) [24]
*Fruhstorferiola* sp.	KU355786	Direct submission
*Fruhstorferiola tonkinensis*	NC_031817	Zhang and Lin (2016) [53]
*Indopodisma kingdoni*	NC_046529	Chang et al. (2020) [24]
*Kingdonella bicollina*	NC_023920	Zhi et al. (2016) [54]
*Kingdonella pienbaensis*	MK903565	Chang et al. (2020) [25]
*Melanoplus bivittatus*	MG993426	Song et al. (2018) [17]
*Melanoplus differentialis*	NC_057646	Direct Submission
*Ognevia longipennis*	NC_013701	Direct Submission
*Paratonkinacris vittifemoralis*	NC_046530	Chang et al. (2020) [24]
*Pedopodisma emeiensis*	NC_046561	Chang et al. (2020) [24]
*Pedopodisma funiushana*	NC_046546	Chang et al. (2020) [24]
*Pedopodisma tsinlingensis*	KX857635	Qiu et al. (2020) [26]
	NC_032303	Direct Submission
*Pedopodisma wudangshanensis*	NC_046547	Chang et al. (2020) [24]
*Prumna arctica*	NC_013835	Sun et al. (2010) [55]
*Qinlingacris elaeodes*	KM363599	Li et al. (2016) [56]
*Qinlingacris taibaiensis*	NC_027187	Direct Submission
*Rhinopodisma eminifrontus*	MK903556	Chang et al. (2020) [25]
*Sinopodisma houshana*	NC_033905	Qiu et al. (2020) [26]
*Sinopodisma lofaoshana*	NC_046562	Chang et al. (2020) [24]
*Sinopodisma lushiensis*	NC_046549	Chang et al. (2020) [24]
*Sinopodisma pieli*	KX857633	Qiu et al. (2020) [26]
	NC_051867	Liu et al. (2017) [18]
*Sinopodisma qinlingensis*	NC_056238	Qiu et al. (2020) [26]
*Sinopodisma rostellocerca*	NC_052716	Chen et al. (2020) [4]
*Sinopodisma wulingshanensis*	NC_033906	Qiu et al. (2020) [26]
*Tonkinacris sinensis*	NC_032716	Zhang et al. (2017) [57]
*Xiangelilacris zhongdianensis*	NC_046533	Chang et al. (2020) [24]
*Yunnanacris wenshanensis*	KX296781	Direct Submission
*Yunnanacris yunnaneus*	NC_030586	Hu et al. (2016) [58]
*Zubovskya koeppeni*	MK903579	Chang et al. (2020) [25]
**Oxyinae**		
*Caryanda elegans*	NC_036750	Yuan et al. (2019) [59]
*Caryanda xinpingensis*	NC_030165	Hu et al. (2017) [60]
*Caryandoides hunanica*	NC_053659	Zeng et al. (2021) [29]
*Fer nigripennis*	NC_053658	Zeng et al. (2021) [29]
*Longchuanacris curvifurculus*	NC_036994	Hu et al. (2018) [61]
*Oxya adentata*	MK903571	Chang et al. (2020) [25]
*Oxya agavisa*	NC_045883	Li et al. (2020) [22]
*Oxya chinensis*	NC_010219	Zhang and Huang (2008) [62]
*Oxya hainanensis*	MN083185	Chang et al. (2020) [24]
	NC_045928	Li et al. (2020) [22]
*Oxya hyla*	NC_032076	Song et al. (2016) [63]
*Oxya hyla intricate*	KP313875	Dong et al. (2016) [64]
*Oxya japonica*	NC_043773	Li et al. (2020) [22]
*Oxytauchira brachyptera*	MT916721	Zeng et al. (2021) [29]
	NC_046570	Chang et al. (2020a) [24]
*Oxytauchira flange*	NC_053745	Zeng et al. (2021) [29]
*Paratoacris reticulipennis*	NC_053660	Zeng et al. (2021) [29]
*Pseudoxya diminuta*	NC_025765	Tang et al. (2014) [65]
**Spathosterninae**		
*Spathosternum nigrotaeniatum*	MG993439	Song et al. (2018) [17]
*Spathosternum prasiniferum sinense*	KM588074	Zhou et al. (2016) [66]
*Spathosternum prasiniferum prasiniferum*	NC_046532	Chang et al. (2020) [24]
**Dericorythidae**		
**Conophyminae**		
*Conophyma zhaosuensis*	ON943040	This study (mt1938)
*Conophymacris viridis*	NC_046528	Chang et al. (2020) [24]
**Dericorythinae**		
*Dericorys annulata*	NC_046555	Chang et al. (2020) [24]
**Pamphagidae**		
**Pamphaginae**		
*Filchnerella kukunoris*	MK903590	Chang et al. (2020) [25]
*Filchnerella yongdengensis*	MK903560	Chang et al. (2020) [25]

**Table 2 insects-14-00085-t002:** Initiation and termination codons of protein coding genes (PCGs) of the newly sequenced complete mitogenomes.

PCGs	Initiation Codons			Termination Codons		
	mt1825	mt1826	mt1938	mt1825	mt1826	mt1938
ND2	ATG	ATG	ATG	TAA	TAA	T
COX1	ACC	ACC	CAA	T	T	T
COX2	ATG	ATG	ATG	TAA	TAA	TAA
ATP8	ATT	ATT	ATC	TAA	TAA	TAA
APT6	GTG	ATG	ATG	TAA	TAA	TAA
COX3	ATG	ATG	ATG	TAA	TAA	TAA
ND3	ATG	ATT	ATT	TAA	TAA	TAA
ND5	ATT	ATT	ATT	TAA	TAA	T
ND4	ATG	ATG	ATG	T	TAA	T
ND4L	ATG	ATG	ATG	TAA	TAA	TAA
ND6	ATG	ATG	ATG	TAA	TAA	TA
CYTB	ATG	ATG	ATG	TA	TAA	TAA
ND1	ATA	ATA	ATG	TAG	TAG	TAG

Note: mt1825: *Longzhouacris mirabilis*; mt1826: *Ranacris albicornis*; mt1938: *Conophyma zhaosuensis*.

**Table 3 insects-14-00085-t003:** Total numbers of different types of base mismatches in tRNAs of the three newly sequenced mitogenomes.

Species	A–A	A–G	A–C	G–U	C–U	U–U
mt1825	1(trnD)	1(trnW)	0	28	0	4(trnQ,trnD,trnH,trnS2)
mt1826	1(trnD)	1(trnW)	0	22	0	4(trnQ,trnE)
mt1938	1(trnD)	1(trnW)	0	16	0	3(trnI, trnQ, trnH)

**Table 4 insects-14-00085-t004:** Distribution of G–U base mismatches in tRNAs of the three newly sequenced mitogenome.

Transfer RNA	mt1825	mt1826	mt1938	Transfer RNA	mt1825	mt1826	mt1938
trnI	1	0	0	trnR	1	1	1
trnQ	1	1	1	trnN	0	0	0
trnM	0	0	0	trnS1	1	1	2
trnW	0	0	0	trnE	1	0	1
trnC	2	1	0	trnF	5	2	2
trnY	2	2	2	trnH	1	3	0
trnL2	1	1	0	trnT	1	1	0
trnD	0	0	0	trnP	2	2	2
trnK	1	1	0	trnS2	1	0	0
trnG	1	1	2	trnL1	1	1	0
trnA	3	2	2	trnV	2	2	1

## Data Availability

The mitogenome sequencesare deposited in GenBank under accession number ON943039 for *Ranacris albicornis*, ON931612 for *Longzhouacris mirabilis* and ON943040 for *Conophyma zhaosuensis*, respectively.

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
