# Peer review of "The Complete Mitogenomes of Three Grasshopper Species with Special Notes on the Phylogenetic Positions of Some Related Genera"

_insects, 2023, doi:10.3390/insects14010085_

Round 1

Reviewer 1 Report

Zhang et al. present a study regarding the phylogenetic position of grasshopper. The whole manuscript is very readable and the results are solid as well. I just suggest improving the resolution of the figure for phylogenetic tree as much as possible.

Author Response

Comment of the reviewer 1: I just suggest improving the resolution of the figure for phylogenetic tree as much as possible.

Response: Our figures for phylogenetic trees have enough resolution as high as 600 dpi. Those the reviewer saw may be impressed by the manuscript submitting system.

Reviewer 2 Report

This paper describes three new mitogenomes of grasshoppers, performs phylognetic reconstructions, makes suggestions for the taxonomy of the subfamilies, and supports these findings with morphological commentary. The findings are extremely clear, and the writing is unimpeachable. I find very few things to comment on, I think this is a necessary and nicely presented study. I only wish the authors had gone further, sequenced more specimens, made stronger claims about the subfamilies, etc, but I understand that some of these things happen over time.

There are several changes I suggest the authors make before final publication:

[1] The introduction focuses mostly on the position of the three newly sequenced species, Longzhouacris, Ranacris, and Conophyma. However the abstract and results introduce new results for other genera, including Emiacris, Oxya, and Choroedocus. I recommend mentioning the findings about the new mitogenomes in the abstract, and giving more background about Emiacris and the other unmentioned genera in the introduction. In general, the narrative could be more clear so that the reader can easily understand the main findings of the paper.

[2] I strongly suggest the authors do not display relationships with very low support in their main figures. For example, in figures 4 and 5, based on a single gene analysis, most relationships have support ~.70 pp. These may be low enough to be not very suggestive of any particular relationship. I think it would be more appropriate to choose a cutoff (e.g. 0.95 posterior, 90 boostrap) and collapse any node below this cutoff to a polytomy. This would make it much more clear how these results should be interpreted (e.g. the non-monophyly of Dericorythidae)

minor suggestions:

[3] line 56 type "It" -> "it"

[4] line 323 "However" should perhaps be changed to "Subsequently", is that right?

[5] Before publication, can the authors state where online their supplemental files will be stored (sorry if I missed this somewhere, the manuscript lists xxx).

Author Response

 Response to reviewer 2:

  1. I recommend mentioning the findings about the new mitogenomes in the abstract, and giving more background about Emeiacris and the other unmentioned genera in the introduction.

Response: The findings about the new mitogenomes have been mentioned briefly in the abstract. More background about Emeiacris, Choroedocus, Conophymacris and Xiangelilacris have been given in the introduction section.

  1. I think it would be more appropriate to choose a cutoff (e.g. 0.95 posterior, 90 boostrap) and collapse any node below this cutoff to a polytomy.

Response: We accepted the reviewer's suggestion and used the cutoffs of 0.95 posterior and 90 bootstrap to collapse the node below this cutoff to a polytomy (see Figure 3 and Supplementary Materials Figures S3).

  1. line 56 type "It" -> "it"

Response: We have corrected the error.

  1. line 323 "However" should perhaps be changed to "Subsequently", is that right?

Response: We have changed "However" to "Subsequently".

  1. Before publication, can the authors state where online their supplemental files will be stored (sorry if I missed this somewhere, the manuscript lists xxx).

Response: We will upload all Supplementary Materials to the submission system, and this online link is usually added by the publisher after the manuscript is accepted.

Reviewer 3 Report

Overall, I found this manuscript to be comprehensive on its subject matter, well-written, and containing information useful to both the entomological and grasshopper taxonomy communities. I endorse this manuscript for publication with minor revisions as indicated in the reviewed version of the manuscript that I've attached. I've included 128 edits/comments that should be addressed before final acceptance. The majority are simply grammatical issues, along with some comments about how to improve the manuscript's content and figures. If addressed, all of these suggested changes will, in my opinion, make this work even stronger.

Author Response

Response to reviewer 3:

According to suggestions in the file "peer-review-25268061.v1.pdf", we have corrected the grammatical error point-by-point. Below are the responses to the comments by the reviewer 3 proposed in the file "peer-review-25268061.v1.pdf".

  1. Line 53-54: note that's good taxonomic practice to include the author name and year in ( ) after the first full use of a species name in a paper.

Response: We have added these information to the manuscript.

  1. Line 92: included a section about how the photographs were taken and the figures were created - for example, what type of cameras, photostacking software, editing, etc.

Response: We have added the methods how the photographs were taken and the figures were created.

  1. Line 105: but at what ethanol %? That's important to know for long-term DNA preservation. Also, room temperature is not optimal, freezing is.

Response: We have clarified this issue in the manuscript.

  1. Line 111: this is pretty outdated, so for at least the Melanoplinae, I suggest reviewing and citing Woller and Song, 2017: DOI 10.1002/jmor.20642.

Response: We have revised the statement and changed the citation to Woller and Song, 2017.

  1. Line 160, 176, 258: formatting issue.

Response: We cannot be able to solve this issue. Please let the type setting team help us resolve this issue.

  1. Line 356: start a new paragraph here.

Response: We have accepted the suggestion.

  1. Line 378: I suggest labeling morphological characters like these in the figures - even for someone like me who knows the characters pretty well can have a hard time finding them in unfamiliar species.

Response: We have added arrows to the corresponding points of the figures (See Figure 5).

  1. Line 381: this is incorrect - the arch of aedeagus is part of the ectophallus since it's attached to the aedeagus (see Woller and Song, 2017: DOI 10.1002/jmor.20642 for good diagrams).

Response: We have revised the statement according to Woller and Song's (2017) reference.

  1. Line 387, 399: new paragraph

Response: We have accepted the suggestion.

  1. Line 401: clarify - do you simply mean including more species?

Response: We have revised the statement to clarify the issue the reviewer pointed out.

  1. Line 416: underline genus names since the rest is italicized already.

Response: We have accepted the suggestion.

  1. Line 419-420: I'm not following this - it was placed into 3 subfamilies at once? Clarify.

Response: It is true that the genus Conophymacris is currently placed into 3 subfamilies at once.

  1. Line 431: add Figure references

Response: We have added the Figure references

  1. Line 433: I'm not seeing the "true arch sclerite" in this pic, which is why I think arrows and labels would help immensely.

Response: We have added arrows to point out where the arch is.

  1. Line 447-448: why is this title not italicized like the rest? italicize genus names.

Response: We have corrected this formatting issue.

  1. Line 628: why is this group of citations all blue?

Response: In the resubmission of the manuscript, the academic editor request us to cite all references where we took mitogenome data. We use blue color to show the added references so that the academic editor can see our correction more easily. Now we have changed the color of this group of citations to black.

Reviewer 4 Report

This is an important study. Zhang et al. sequenced, assembled, and annotated the mitochondrial genome of three grasshopper species (Longzhouacris mirabilis, Ranacris albicornis, and Conophyma zhaosuensis). This work is useful and has provided valuable information about phylogenetic analyses and evolutionary studies. I would suggest some revisions as follows:

Major comments:

The three mitogenome sequences (ON943039, ON931612, and ON943040) generated by this study are not publicly available. The results, including phylogenetic trees, can't be assessed without having these mitogenome sequences. Can you please make these sequences accessible?  

Minor comments:

1- Line 89: It is best to avoid referring to tables in the introduction.

2- Line 108-109: please merge this sentence with the next sentence to represent a paragraph. 

3- The first mention of figures 6 and 7 in the discussion section is inappropriate, and figures should be previously presented in the results section.

4- I suggest moving tables 2 and 3 to supplementary.

5- I suggest moving figures 3 and 5 to supplementary.

6- Line 232: remove the space after the word "genes".

7- I would suggest improving the quality of figures 2, 3, 4, and 5 to increase the readability.  

Author Response

Response to reviewer 4:

Major comments: The three mitogenome sequences (ON943039, ON931612, and ON943040) generated by this study are not publicly available. The results, including phylogenetic trees, can't be assessed without having these mitogenome sequences. Can you please make these sequences accessible?

Response: We have contact the NCBI to release these data.

Minor comments:

  1. Line 89: It is best to avoid referring to tables in the introduction.

Response: We have deleted the referring to table 1 in the introduction section.

  1. Line 108-109: please merge this sentence with the next sentence to represent a paragraph.

Response: We have merged these two sentenses into a paragraph.

  1. The first mention of figures 6 and 7 in the discussion section is inappropriate, and figures should be previously presented in the results section.

Response: Although the reviewer's opinion is reasonable, We think it does not matter because the morphological characters of a known species can be regarded as a kind of known informative data for an experienced taxonomist (just like the information in the references) and we just displayed (visualized) the known evidence via the photos we took ourself. So it is not necessary to previously present the morphological figures in the result section since the main analysis of our manuscript is molecular phylogenetics.

4- I suggest moving tables 2 and 3 to supplementary. 

Response: We have moved tables 2 and 3 to supplementary.

5- I suggest moving figures 3 and 5 to supplementary. 

Response: We have moved figures 3 and 5 to supplementary.

6- Line 232: remove the space after the word "genes".

Response: We have deleted the redundant space.

7- I would suggest improving the quality of figures 2, 3, 4, and 5 to increase the readability.  

Response: Our figures for phylogenetic trees have enough resolution as high as 600 dpi. Those the reviewer saw may be impressed by the manuscript submitting system.

Round 2

Reviewer 4 Report

The authors have addressed my comments and I am satisfied with the improvements. I recommend publication in the journal.